# The Role of Beetroot Ingredients in the Prevention of Alzheimer's Disease

**Julian Szymański \***, **Dominik Szwajgier \*** and **Ewa Baranowska-Wójcik**

Department of Biotechnology, Microbiology and Human Nutrition, University of Life Sciences, Skromna Street 8, 20-704 Lublin, Poland

\* Correspondence: julian.szymanski@up.lublin.pl (J.S.); dominik.szwajgier@up.lublin.pl (D.S.)

**Abstract:** Beets (*Beta vulgaris* L.) are a source of numerous bioactive compounds, including betalain pigments, phenols, and saponins. The bioactive compounds show neuroprotective properties due to their antioxidant activity (they protect cells against oxidative stress caused by the overaccumulation of reactive oxygen species), anti-inflammatory effects, and the ability to lower the activity of acetylcholinesterase. The most common pigment present in beetroot is betanin. Scientists have repeatedly demonstrated the antioxidant activity of this compound, which is capable of protecting the cell membrane of neurons against peroxidation. The phenolic compounds present in the root showed the same effect. Phenolic acids are the most prevalent group of such compounds, including gallic, ferulic, and vanillic acids. It has been shown that neurodegenerative diseases induced artificially (e.g., with sodium fluoride or trimethyltin chloride) are reversed by the administration of betanin. A similar protective effect has been demonstrated in streptozotocin-induced disease models. For example, the administration of vanillic acid improved spatial learning ability. Hence, processed beetroot (juices, jams, etc.) can offer viable benefits in preventing neurodegenerative diseases such as Alzheimer's disease (AD). The following review presents a detailed summary of the current state of knowledge regarding the most important bioactive compounds present in beetroots and their applicability in AD prevention and support therapy.

**Keywords:** beetroot; neurodegenerative diseases; Alzheimer's; oxidative stress





## 1. Introduction

Our contemporary world is conducive to neurodegenerative diseases. This is due to the growing pace and stressfulness of our lives, as well as the continuously increasing average life expectancy [1,2]. We often fail to pay sufficient attention to what we do and do not eat. Generally speaking, neurodegenerative diseases can be characterized as a group of nervous system disorders leading to impairment of neural function. They are most commonly diagnosed in the elderly, but also increasingly occur in younger patients who can be affected in their middle age or even youth. The most common neurodegenerative disease is Alzheimer's disease (AD). The clinical symptoms depend on the specific type of condition, but most entail disorders in terms of memory—remembering—and motor functions—mobility and coordination problems [3].

The difficulty in treating neurodegenerative diseases is caused by the limited ability to deliver pharmaceuticals to the brain. This is due to the presence of a specific blood–brain barrier that reduces and limits the bioavailability of the administered drugs. Disturbances in the proper functioning of the antioxidant system cause an increase in oxidative stress, which can lead to the development of neurodegenerative diseases. Various researchers have suggested the viability of prophylaxis in the form of externally provided ingredients with proven antioxidant properties. In this context, compounds that are readily available and have no adverse health effects are particularly sought after. Food products, e.g., fruits and vegetables, can be a valuable source of antioxidants that reduce oxidative stress, which

occurs when the antioxidative defense mechanisms prove insufficient due to overexposure to reactive oxygen species. The oxidative balance can be upset due to the excessive presence of toxic peroxides and free radicals, leading to oxidative cell damage, including to key cell components such as proteins, lipids, and genetic material. Chemically, oxidative stress can be described as a reduction in the reduction capacity of cellular redox cells, such as glutathione, resulting in cell damage [2,4]. Chemically, oxidative stress can be described as a reduction in the reduction capacity of cellular redox cells, such as glutathione, resulting in cell damage. Oxidative stress is a pathological cellular condition characterized by an imbalance between the production and inactivation of free radicals. Oxidative stress is conducive to the emergence of neurodegenerative diseases. It triggers multidirectional changes in the affected organ. Therefore, it can be posited that oxidative stress can significantly contribute to the onset of diseases such as AD [2,5]. Reactive oxygen species such as superoxide anion, hydroxyl radical, and hydrogen peroxide are present in the human brain [6]. It should be noted that the brain is a very delicate and sensitive organ. It is very susceptible to oxidative changes. Sixty percent of the brain consists of lipids, which are sensitive to the peroxidation process when exposed to reactive oxygen species [7]. The function of protecting neurons against reactive oxygen species is performed by astrocytes, the largest glial cells of the central nervous system. They regulate the redox balance through the accumulation and release of endogenous antioxidants—ascorbic acid and glutathione. The brain tissue is predisposed to store metal cations such as copper and iron. Metals catalyze the so-called Fenton reaction, which produces a reactive oxygen species—the hydroxyl radical, which can damage neurotransmitters and, as a consequence, cause central nervous system impairment [4]. Cellular respiration takes place in the mitochondria, which are necessary for the proper functioning of cells. Damage to them caused by oxidative stress can precipitate the development of neurodegenerative diseases. The function of the respiratory chain and oxidative phosphorylation is to facilitate oxidation of flavin adenine dinucleotide and nicotinamide adenine dinucleotide, and consequently the production of energy in the form of adenosine-5'-triphosphate. Reactive oxygen species cause damage to mitochondrial DNA, and consequently to mitochondrial proteins, which reduces the amount of ATP produced and triggers cell apoptosis [5].

## 2. General Characteristics of Alzheimer's Disease

AD is among the most common conditions classified as neurodegenerative. It is an insidious disease developing over the course of many years without acute symptoms. When the early symptoms are observed, they are often ignored and attributed to age or stress [1,8]. Based on the bulk of current medical knowledge on this topic, AD is classified as a progressive and incurable disease. It is manifested by dementia, and as such is a condition that significantly hinders the patients' ability to function normally by affecting their capacity for rational thought and remembering. It also negatively affects communicative abilities and food ingestion (swallowing reflex) [9]. A number of hypotheses as to the causes of the disease are currently explored, in particular: the cholinergic hypothesis, amyloid hypothesis, and tau hypothesis. However, there are also others that link the development of the disease to biochemical disorders related to neurotransmitters, infectious stimuli, genetic factors associated with gene defects, or increased levels of oxidative stress [10,11]. One posited explanation regarding the etiology of the disease is the cholinergic hypothesis that pertains to the transmission of neural impulses within the cholinergic system. The system is involved in the regulation of higher cortical functions, specifically the ability to concentrate and focus one's attention on particular tasks, as well as the capacity to learn and remember [12]. It is hypothesized that AD may be caused by the insufficient concentration of the neurotransmitter acetylcholine in the central nervous system. To improve the efficiency of neurotransmission, therapeutics stimulating adequate (nicotinic and muscarinic) receptors are used alongside blockers of acetylcholinesterase, the enzyme responsible for acetylcholine hydrolysis [13]. The amyloid hypothesis posits that AD is in fact amyloidosis and explains that the direct cause of the disease is excessive, pathological accumulation

of amyloid proteins. The cause of the disease is related to the gene encoding proteins present in the cytoplasmic membrane of neurons. The gene is located on chromosome 21 and encodes a peptide composed of approx. 700 amino acids, the so-called β-amyloid precursor protein [11,14,15]. Under physiological conditions, non-amyloidogenic cleavage of beta-amyloid precursor protein occurs with the participation of α-secretase. As a result, a soluble form of the protein is formed. In amyloidogenic conditions, the beta-amyloid precursor protein is cleaved by β-secretase and γ-secretase. The final products of these transformations are insoluble, toxic β-amyloid polypeptides containing 40 or 42 amino acids. They are characterized by a higher predisposition for the formation of larger amyloid structures. Due to the extracellular emergence and accumulation of β-amyloid protein, so-called amyloid plaques (senile plaques) are formed. Amyloid plaques bind with neuronal receptors and modify the structure of intraneuronal connections, disrupting the transfer of neural impulses [16]. Yet another hypothesis aiming to explain the onset of AD connects it to excessive phosphorylation of the tau protein. Tau proteins are related to the microtubules responsible for the correct neuronal cytoskeleton. Tau proteins contribute to stabilizing the structures responsible for the formation of the cytoskeleton. They maintain the correct distance between microtubules. Physiologically, tau proteins are present in six distinct forms. All those forms are encoded by the same gene (MAPT), located on chromosome 17. The differences between the respective forms of tau proteins result from varying combinations of encoding sequences (exons). MAPT mutation leads to the formation of tau proteins with excessive phosphorylation. The structure of proteins produced due to the transcription of the faulty gene differs from that of the physiological protein. As a result, such proteins begin to bind with each other to form larger pathological structures, referred to as neurofibrillary tangles, that accumulate in the cytoplasm of nerve cells. This, in turn, disrupts axonal transport, and consequently leads to neuron death [17–19].

### 2.1. Neuroprotective Properties of Beets in AD

Treatment of neurodegenerative diseases relies primarily on pharmacotherapy. In recent years, however, increasing emphasis has also been placed on choosing the proper diet to complement and facilitate the therapy. Some compounds present in plant tissues have been shown to have properties that can slow down the progression of such diseases [20].

Currently, a major trend in scientific research focuses on identifying valuable natural compounds of plant origin. In this context, the beet (*Beta vulgaris* L.) proves to be a valid contender as it is rich in numerous bioactive compounds, including ones with proven antioxidant properties, that may have a positive impact on the condition of human organs [21–23].

Our knowledge as to the impact of consuming beetroots, or preserves made therefrom, on the prevention of various diseases, including neurodegenerative ones, has improved significantly over the last dozen or so years. Wootton-Beard and Ryan [24] demonstrated that beetroot juice, digested under in vitro conditions, showed antioxidant activity (FRAP method) and a high content of total phenols. The antioxidant capacity of the juice before digestion and after the process of simulated digestion in the stomach and in the colon was, respectively, 697.9 ± 1.6 μmol/70 mL, 2361.2 ± 20.9 μmol/70 mL, and 1740.3 ± 21.1 μmol/70 mL, expressed as the equivalent of iron (II) sulfate (VI). Another team of scientists confirmed that the consumption of beetroot may improve cerebral blood flow, and consequently cognitive function [25]. Wightman et al. [26] observed improved cerebral blood flow parameters induced by the consumption of beetroot by adult subjects, which also triggered an improvement in cognitive functions. Sulakhiya et al. [27] analyzed the impact of beet leaf extract in cases of anxiety, depression, and oxidative stress in mice induced by acute restraint stress. The therapeutic dose administered orally was 100 and 200 mg of ethanol beet leaf extract per kg b.w. Based on the conducted analyses, it was concluded that the beet extract showed anti-anxiety and antidepressant properties. Mokhtari et al. [28] studied the impact of beet leaf extract on the inhibition of AD progression. The experiment was conducted on rats with induced AD receiving a therapeutic dose of the beet

extract (100 and 200 mg/kg b.w.). Based on the conducted analyses, the authors concluded that a 15-day therapy with the extract can reduce learning and memory disorders related to AD. Olasehinde et al. [29] conducted an experiment with a view to determining the impact of beet consumption on the cognitive functions and the overall progression of AD in rats receiving scopolamine. The rats received beetroot powder as a food supplement in a dose of 2 and 4%, which was observed to improve the animals' cognitive functions. A research team lead by Rehman [30] analyzed beetroot extract in terms of its ability to inhibit the activity of acetylcholinesterase. The studied extract (100 µg dry matter per mL) showed the capacity to lower the activity of acetylcholinesterase (93.3%) in comparison to the standard drug donepezil (94.2% of enzyme inhibition when used in the same concentration). Aliahmadi et al. [31] conducted a study in a group of patients suffering from type II diabetes. The aim of the study was to determine the impact of consuming raw beet (100 g a day for 8 weeks) on the patients' cognitive functions. The latter were assessed using the Toulouse-Pieron (TP) and Digit Learning (DL) tests. It was found that the consumption of raw beet significantly improved antioxidant activity as well as cognitive functions. Ertas et al. [32] studied the impact of beet therapy in cases of cognitive disorders, reduced cholinergic transmission, and neuroinflammation induced by the administration of streptozotocin into the rats' cerebral ventricles. The therapeutic dose of beet leaf extract was 2 g/kg b.w. for 21 days. Based on the conducted experiment, it was concluded that beet leaf extract improved the animals' cognitive functions and memory. The effects could be attributed to the observed anticholinesterase and anti-inflammatory properties of the extract. The administration of beet leaf extract significantly reduced the activity of acetylcholinesterase in the hippocampus and parts of the cerebral cortex in rats receiving streptozotocin, respectively by 40% and 30% relative to the control. Shaban et al. [33] analyzed the neuroprotective properties of beet juice in rats with lead-induced neurotoxicity (Pb acetate, PbAc, dosed at 40 mg/kg b.w.). The administration of lead intensified lipid oxidation, lowered glutathione levels in cerebral tissues, and lowered antioxidant capacity. In order to verify the actual effectiveness of the beet juice therapy, for comparison, one test group received a standard medicine (DMSA, dimercaptosuccinic acid, 50 mg/kg b.w.) and a combination of beet juice and DMSA. Based on the conducted study, it was observed that the beet juice increased the level of glutathione (GSH) compared to the group receiving Pb by, respectively, up to approx. 25 mg GSH/g of cerebral tissue and approx. 15 mg GSH/g of cerebral tissue. The therapeutic dose of beet juice was 8 mL/kg of body weight. The combined use of juice and DMSA as well as DMSA alone returned slightly higher glutathione levels as compared to the application of just the juice (approx. 30 mg GSH/g cerebral tissue). The therapy additionally improved the antioxidant capacity of brain tissues. The analysis was conducted in a group of animals receiving a toxic dose of PbAc, a group receiving beet juice, and a group receiving beet juice in combination with DMSA. The results reported for the above groups of animals were as follows: approx. 0.4, 0.75, and 0.7 µg BHT/g cerebral tissue, respectively. Administration of just the standard drug yielded results similar to those observed for the combination of DMSA and juice. The team analyzed the activity of acetylcholinesterase, a key enzyme involved in AD pathogenesis. The use of the therapeutic dose of beet juice lowered the enzyme's activity in cerebral tissue by approximately 15%, compared to the group receiving PbAc. By comparison, the use of the juice in combination with the standard drug DMSA yielded even better results in terms of lowering acetylcholinesterase activity (approx. 40%). Based on the conducted analyses, it was concluded that beet juice may constitute an effective neuroprotective agent in cases of lead poisoning. The use of the juice in combination with DMSA reduced oxidative stress, inflammation, and harmful lead levels in cerebral tissues. Wulandari et al. [34] conducted a study in a group of 25 farmers who consumed a beet-based beverage for a period of two months (1 dm$^3$ daily; the beverage was prepared using 150 mg of beets and 1000 mL of water). Each farmer consumed the juice in two daily helpings (2 × 500 mL). The activity of acetylcholinesterase in the blood was measured prior to the experiment and after 2 months of daily consumption. In 12 of

the subjects, the activity of acetylcholinesterase, the enzyme responsible for acetylcholine hydrolysis, was reduced.

### 2.2. Individual Bioactive Compounds from Beetroots in the Prevention and Treatment of AD Nitric Pigments

Beets are a source of specific nitric pigments in the form of betalamic acid derivatives containing nitrogen. Two distinct groups of betalain pigments can be distinguished—betacyanins (red-violet color) and betaxanthins (orange-yellow color). Examples of betacyanins include betanin, isobetanin, and neobetanin, while betaxanthins include vulgaxanthin I, vulgaxanthin II, and indixanthin [22]. Betalain pigments are the compounds responsible for the characteristic color of beetroots. The most popular betacyanin present in beets is betanin (betanidin 5-O-D-glucoside). It is the only one approved for use in food, cosmetics, and pharmaceuticals [35]. Figure 1. presents a breakdown of the main pigments present in beetroots. In the opinion of Mikołajczyk-Bator [36], it is due to the presence of those betalain pigments that beets show such strong antioxidant properties. Indeed, betalain pigments have been analyzed in numerous studies in terms of such effects. Gengatharan et al. [37] described betalain pigments as natural pigments that may be viably used in the production of functional food. Wootton-Beard and Rayan [24] reported that betanin shows strong antioxidant properties, and Kathiravan et al. [38] verified that betanin has antioxidant effects as it protects lipids present in cellular membranes against peroxidation. Cai et al. [39] confirmed, in a test using ABTS and DPPH radicals (TEAC equivalent antioxidant capacity), that the antioxidant capacity of betalain pigments was, respectively, 7.5 and 3.0 times higher than in the case of vitamin C. Vulić et al. [40] analyzed beet pulp from five plant cultivars in terms of antioxidant activity. Based on the recorded results, they concluded that the highest antioxidant capacity was observed for the Detroit cultivar. The results were confirmed using the DPPH free radical method (EC50 = 2.06 $\pm$ 0.10 µg/mL). Hacioglu et al. [41] conducted an experiment to determine the neuroprotective properties of betanin in the context of neurotoxicity induced in rats using sodium fluoride (80 mg/L water). The therapeutic doses of betanin were 0.25, 0.5, and 1 mM. The results revealed that sodium fluoride induced a decrease in glutathione levels (GSH) in synaptosomes, as compared to the control, from 0.57 $\pm$ 0.11 to 0.31 $\pm$ 0.34 µmol GSH/mg protein. It was further observed that all three of the betanin doses showed neuroprotective effects, but the best results in terms of improving GSH levels were recorded for the 0.5 mM dose (0.54 $\pm$ 0.11 µmol/mg protein). The applied toxic dose of sodium fluoride increased the oxidation of lipids on the synaptosomes, and consequently the levels of malondialdehyde (MDA)—the marker of lipid peroxidation. The level of synaptosomal MDA in the control group and the group receiving sodium fluoride was, respectively, 5.76 $\pm$ 1.53 and 8.21 $\pm$ 2.68 µmol/mg of protein. All the therapeutic doses of betanin induced a reduction in the values of synaptosomal MDA relative to the group receiving only sodium fluoride. The best effects were reported for the 1 mM dose (5.97 $\pm$ 1.26 µmol MDA/mg protein). The toxic dose of sodium fluoride reduced catalase activity by approx. 11 units/mg of protein relative to the control group's 17.53 $\pm$ 2.44 units/mg of protein. All betanin doses increased the activity of the enzyme, and the best results were recorded for the dose of 1 mM (17.81 $\pm$ 2.35 units/mg of protein). However, betanin doses of 0.25 and 0.5 mM also increased the catalase activity, respectively to 12.25 $\pm$ 1.61 and 17.28 $\pm$ 1.89 units/mg of protein. On the basis of the recorded results, the researchers concluded that betanin shows neuroprotective activity in cases of neurotoxicity induced by fluorine compounds. Motawi et al. [42] analyzed the neuroprotective activity of betanin against stress caused by the overdose of paracetamol (400 mg/kg) and diclofenac (10 mg/kg) in rats. In the group of animals receiving paracetamol and diclofenac as well as betanin supplementation (25 mg/kg b.w.), the authors observed a lessening of oxidative stress and a reduction of most biochemical and histopathological changes due to the excessive consumption of paracetamol and diclofenac. The eventual conclusion was that betanin may serve as an effective neuroprotective agent. Thong-Asa et al. [43] conducted a study in a group of mice intraperitoneally administered with trimethyltin chloride (single intraperitoneal injection

of 2.6 mg/kg b.w.) to induce a neurodegenerative disorder. The therapeutic dose of betanin dissolved in normal saline was administered orally at 50 and 100 mg/kg b.w. The authors observed that when dosed at 100 mg/kg b.w., betanin showed neuroprotective activity against the neurodegenerative condition induced with trimethyltin chloride. Imamura et al. [44] studied the impact of betalain pigments present in beet plants on the process of β-amyloid aggregation, i.e., one of the causes of AD. The conducted analyses showed inhibition of β-amyloid aggregation, which confirmed that betalain pigments may provide an effective tool in the fight against AD. The hypotheses were corroborated with the use of the thioflavin T fluorescence test, circular dichroism spectroscopy, and electron microscopy observations (TEM). Thioflavin T is a fluorescent pigment that binds with β-amyloid. As a result, the fluorescence level is increased. The betalain pigments were used in a range of concentrations (6.25, 12.5, 25, and 50 μM). For each concentration, reduced fluorescence levels were observed, which suggests that the pigments inhibited the process of forming pathological β-amyloid structures. Elkewawy and Elbadrawy [45] evaluated the protective activity of beetroot juice against neurotoxic aluminum in the brains of male rats (the toxic dose of 20 mg/kg b.w. of aluminum chloride was administered intraperitoneally, once daily, from the 14th day of the experiment until its conclusion). The authors observed that administration of beetroot juice dosed at 15 mL/kg b.w. via a gastric tube, once daily over a period of 6 weeks, yielded positive effects in terms of protection against the oxidative stress induced by aluminum chloride. The level of GSH in the brain was increased after the application of the therapeutic juice dose, as compared to the group receiving only the toxic aluminum dose and was, respectively, $21.74 \pm 1.18$ and $11.14 \pm 0.81$ mg GSH/g of cerebral tissue. The activity of acetylcholinesterase (AchE) in the brain was reduced after administering the juice to $428.60 \pm 23.16$ from $593.80 \pm 30.07$ units AchE/g of cerebral tissue, relative to the group receiving only the toxic dose of aluminum. Shunan et al. [46] conducted a study on rats with a view to determining the protective potential of betanin against AD induced with aluminum chloride (administered orally, 100 mg/kg b.w.). Aluminum is a strong neurotoxin triggering oxidative stress, one of the causes of neurodegenerative diseases. The tested rats received oral doses of betanin (10 and 20 mg/kg b.w.) over a period of 4 weeks. Betanin was reported to have likely reduced learning disorders and caused tissue damage and cholinergic disfunction by inhibiting oxidation. Both of the betanin doses tested reduced the activity of acetylcholinesterase relative to the group receiving only the toxic dose of aluminum chloride. An increase in glutathione level was also observed relative to the group receiving aluminum chloride. The researchers concluded that betanin may be a promising bioactive compound in the pharmacotherapy of neurodegenerative diseases such as AD. Ahmadi et al. [47] studied the anti-inflammatory activity of betanin in simulated microglia cells in terms of the potential to reduce neuroinflammation. As follows from the analysis results, betanin concentrated at 500 μM inhibited liposaccharide (LPS)-induced production of inflammation mediators such as tumor necrosis factor (TNF-α), interleukin 1β (IL-1β), and interleukin 6 (IL-6).

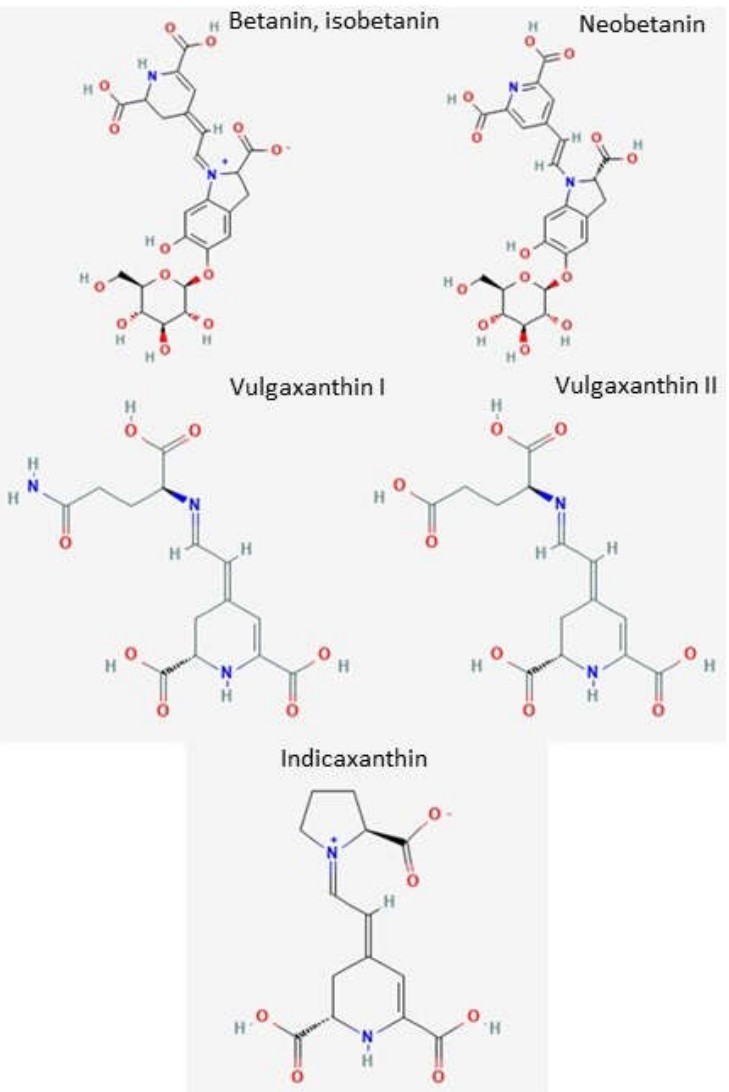

**Figure 1.** Betalain pigments in beetroots (structural formula).

### 3. Phenolic Compounds

The next important group of bioactive compounds present in beetroots are phenols. The group includes flavonoids, phenolic acids, and phenolic amides. The compounds are beneficial to health as antioxidants neutralizing reactive oxygen species. As such, they protect lipids present in cellular membranes against oxidation. It has been reported that consuming food containing flavonoids may slow down the progression of neurodegenerative diseases [48]. The phenolic compounds detected in beetroots are listed in Table 1.

**Table 1.** Phenolic compounds in beetroots.

| Name of Compound | Reference |
| --- | --- |
| Ferulic acid | [22,49] |
| 5,50,6,60-tetrahydroxy-3,30-biindolyl; 5,6-dihydroxyindole carboxylic acid dimer; phenol amides—N-trans-feruloyltyramine; N-trans-feruloylohomovanillylamine | [35] |
| Phenolic acids: caffeic, vanillic, p-coumaric, protocatechuic | [50] |
| Betagarin, vanillic acid, betavulgarin, p-coumaric acid | [51] |
| Gallic acid | [52–54] |
| Betagarin, cochliofilin A, betavulgarin, dihydroisorhamnetin | [40] |

Kathiravan et al. [38] concluded that the total content of phenolic acids was 50–60 µmol/g of dry matter. Several research teams have reported that gallic acid was the most prevalent phenolic compound present in beetroot [52–54]. Nemzer et al. [35] concluded that beetroot contains 5,50,6,60-tetrahydroxy-3,30-biindolil and 5,6-dihydroxyindole carboxylic acid dimer. Additionally, they detected the presence of phenolic amines—N-trans-feruloyltyramine and N-trans-feruloylohomovanillylamine. Maraie et al. [50] reported the presence of phenolic acids, including caffeic, vanillic, p-coumaric, and protocatechuic. Other authors also confirmed the presence of phenolic acids in beetroot extract (ferulic, p-hydroxybenzoic, and protocatechuic acids at, respectively, 132.52 mg, 1.13 mg, and 5.42 mg 100 g per dry matter) [22]. Baiao et al. [51] demonstrated that phenolic compounds present in beetroots, such as betagarin, vanillic acid, betavulgarin, and p-coumaric acid, show considerable antioxidant potential. Compounds with such properties can actively protect cellular membrane lipids against peroxidation. Gallic acid has a wide spectrum of biological properties, including the ability to alleviate inflammation related to oxidative stress [52–54]. Vanillic acid has anti-inflammatory and antioxidant properties, which may allow it to alleviate neuroinflammatory processes. Sharma et al. [55] reported that vanillic acid can ease the symptoms of neurodegenerative diseases caused by oxidative stress and accumulation of β-amyloid. Singh et al. [56] conducted a study with a view to determining the neuroprotective properties of vanillic acid against a neurodegenerative disease induced in mice by intracortical injection of streptozotocin dosed at 3 mg/kg b.w. on the 14th and 16th day. Three experimental doses of vanillic acid were administered orally for 28 days (25, 50, and 100 mg/kg b.w.). Vanillic acid was found to improve the capacity for spatial learning, as confirmed in tests, relative to the control group. Dosed at 25, 50, and 100 mg/kg b.w., vanillic acid lowered the activity of acetylcholinesterase. The results were, respectively, $18.87 \pm 0.39$, $16.92 \pm 0.29$, and $15.52 \pm 0.23$ µmol/min/mg relative to the group receiving only the toxic dose of streptozotocin ($22.34 \pm 0.41$ µmol/min/mg). Based on the recorded results, the authors concluded that vanillic acid may constitute a viable therapeutic agent in the prevention of AD. The antioxidant properties of ferulic acid have also been confirmed, which means that it can potentially have a protective effect on cellular membranes. The acid may also inhibit the development of certain diseases, including neurodegenerative diseases, cardiovascular conditions, and cancer [51,57]. Zhao and Moghadasian [48] studied the content of ferulic acid in beetroot. They reported the result of 25 mg 100/g d.m. A higher ferulic acid content was observed in beetroot pomace (132.52 mg 100/g d.m.) [22]. Singh et al. [58] reported that ferulic acid shows considerable health benefits, including in terms of antioxidant, antidiabetic, anticancer, anti-inflammatory, and neuroprotective properties, with no evident side effects. The neuroprotective capacity of ferulic acid results from its antioxidant properties and the ability to significantly modify the kinetics of β-amyloid aggregate formation (anti-amyloidogenic activity), which is among the causes of AD progression [59]. Ferulic acid shows neuroprotective activity in the course of AD due to its anti-amyloid properties. It inhibits the formation and accumulation of β-amyloid concrements. It also shows the capacity to inhibit β-secretase, an enzyme playing a key role in the etiology of AD [60–62]. Wang [63] reported that ferulic acid can reduce the expression of the proteins responsible for the development of AD, such as tau protein and β-amyloid precursor protein. It was demonstrated that ferulic acid acts anti-inflammatorily by lowering the expression of TNF-$\alpha$, IL-6, and IL-1β [64]. Mori et al. [65] investigated the therapeutic effects of octyl gallate and ferulic acid in a mouse model of AD. Octyl gallate and ferulic acid were dissolved in distilled water with the addition of dimethyl sulfoxide. The mice received octyl gallate, ferulic acid, and octyl gallate + ferulic acid orally, dosed at 30 mg/kg b.w., once daily over a period of 3 months. The control group received distilled water with dimethyl sulfoxide. After the experimental period, it was observed that the therapy improved the animals' cognitive functions, measured using the Y labyrinth and water labyrinth tests. A reduction of pathological β-amyloid concrements in the brain parenchyma was reported. The authors also observed lower activity of β-secretase. Combined therapy with the use of octyl gallate and ferulic acid reduced inflammation and oxidative stress in the nervous system. Szwajgier et al. [66] analyzed the antioxidant activity of selected phenolic compounds. The

conducted experiments revealed that caffeic, protocatechuic, and ferulic acid all showed antioxidant properties measured at, respectively, 1.410, 1.920, and 1.360 mmol/L TEAC (value expressed as Trolox equivalent). Albasher et al. [67] conducted an experiment to determine the potential neuroprotective effects of phenol-rich, methanol beetroot extract in cases of cortical damage induced using the organophosphate pesticide chlorpyrifos. The research team analyzed the content of the phenolic extract and confirmed the presence of phenolic compounds including cinnamic, gallic, vanillic, caffeic, coumaric, and ferulic acid, as well as catechin. The content of total phenols was 91.6 +/− 6.36 mg of gallic acid equivalent in 1 g of fresh material. The content of total flavonoids in beetroot was 112.3 +/− 13.21 mg of quercetin equivalent per gram of fresh material. The experimental group was composed of 28 rats. The animals administered with chlorpyrifos showed a reduced activity of cortical acetylcholinesterase. The neurotoxic effects of chlorpyrifos intensified the oxidation of lipids constituting the cellular walls of neurons. In the group of rats receiving chlorpyrifos (10 mg/kg b.w. once daily for 28 days), the authors reported that the activity of cortical acetylcholinesterase was reduced by approximately half as compared to the control group receiving a saline solution (0.9% NaCl). In the group receiving chlorpyrifos and a beetroot extract (300 µg of extract/kg b.w. for 28 days, once daily an hour before administering 10 mg/kg b.w. of chlorpyrifos), the activity of acetylcholinesterase was observed to be considerably increased, by approximately half when compared to the group receiving chlorpyrifos.

### 4. Saponins

Saponins are a group of chemical compounds found in a variety of plant species. They share a precursor composed of 30 carbon atoms (oxidosqualene). Chemically, they are classified as glycosides. They comprise two parts. Aglycone or the non-sugar part, and glycon—the sugar part. Depending on the character of the aglycone, two classes of saponins are distinguished: triterpenoid saponins and steroid saponins [68]. The presence of bioactive triterpenoid saponins was reported in beetroot, with oleanolic acid and its derivatives constituting the non-sugar part. For instance, Mroczek et al. [69] isolated betavulgarosides I–VIII. Mikołajczyk-Bator et al. [70] analyzed the biochemical composition of several beet cultivars. They identified 44 saponins (Nochowski cultivar). The respective aglycones included oleanolic acid, hederagenin, akebonoic acid, and gypsogenin. In most cases, the non-sugar part was composed of oleanolic acid. Triterpenoid saponins contain one or two sugar radicals bonded with the aglycon by a glycoside bond. The sugar part is usually composed of hexoses, pentoses, 6-deoxyhexoses, and uronic acids. Swenty-seven saponin derivatives of the same have been identified. The remaining saponins were derivatives of akebonic acid (10 saponins), hederagenin (6 saponins), and gypsogenin (1 saponin). In another paper, Mikołajczyk-Bator et al. [71] confirmed the presence of 44 saponins in beetroot, as well as the presence of betavulgarosides I–VIII, which were also reported by Mroczek et al. [68]. Mroczek et al. [72] analyzed extracts from beet leaves and identified 11 saponins with oleanolic acid constituting the aglycone and hexoses, pentoses, and uronic acid as the glycon. The research team quantitatively measured the content of saponins in extracts from the leaves and roots of four beet cultivars—Red Spher, Forono, Egyptian, and Round Dark Red. The content of saponins in the leaf extract was between 5905.6 µg/g of dry matter in Forono and 11,555.9 µg/g d.m. in Red Sphere plants. The content of saponins in beetroot extracts was between 4109.7 µg/g d.m. (Forono) and 16,363.4 µg/g d.m. (Egyptian).

In plants, saponins provide protection against pathogens. Moreover, due to their biological activity, they show anti-inflammatory, antibacterial, anticancer, immunomodulatory, and antiparasitic properties [72]. Unfortunately, no scientific reports are available regarding the use of beetroot saponins in the treatment of neurodegenerative diseases. Nonetheless, attempts have been made to use saponins obtained from other plants in the prevention of such diseases. For instance, Khalil et al. [73] analyzed the therapeutic effect of saponins in AD. Saponins were extracted from the seeds of fenugreek (*T. foenum-graecum* L.). A

methanol extract was used in the study conducted on rats with induced AD (drinking water containing aluminum chloride (III) (0.3%) for 45 days). The therapeutic dose consisted of a methanol extract from fenugreek seeds (0.05, 0.1, and 2% in drinking water for 45 days). A decrease in the activity of acetylcholinesterase, a key enzyme in the etiology of AD, was observed. The AChE inhibitive activity was expressed in the form of the IC50 index. The value of IC50 for the control group receiving only water was $8.2 \pm 2.3$ µg/mL. For the AD group receiving only $AlCl_3$, it was $13.6 \pm 3.2$ µg/mL. The value of IC50 for the study group receiving 0.05, 0.1, and 2% of the methanol saponin extract was, respectively, $36.2 \pm 7.6$, $64.1 \pm 9.2$, and $85.6 \pm 8.4$ µg/mL. On average, the value of inhibitive activity index increased 4.5, 8, and 10.6-fold in the cerebral tissue of rats, relative to the control group receiving only water. By comparison, in the group receiving the reference drug—rivastigmine (0.3 mg/kg b.w. in drinking water for 45 days), the IC50 index was $62.3 \pm 7.7$ µg/mL.

## 5. Changes in the Content of Bioactive Compounds in Beetroots during Processing

Vegetables, including beetroot, are often processed before consumption. This includes a number of technological processes such as thermal and mechanical processing, or lactic fermentation. Processing can impact the content of bioactive compounds in plant tissues, as well as the anti-oxidative capacity of the final product and its respective bioactive ingredients [74]. It is therefore important to consider the changes affecting bioactive compounds during beetroot processing.

Szwajgier et al. [75] observed that high pressure processing of 100% beetroot and carrot juice (80/20 $v/v$, 600 MPa, 15 minutes, at 55 and 65 °C) did not statistically significantly ($p < 0.05$) influence the content of total phenols. High pressure processing at 65 °C caused an increase in the content of betalain pigments (isobetalain, decarboxybetalain, and betalain by, respectively, 100, 14, and 10%). Under the same conditions, the content of vulgaxanthin I was decreased by 65%. Sawicki et al. [76] analyzed the impact of boiling, roasting, and lactic fermentation on the content of nine phenolic acids in beetroot (primarily ferulic and p-coumaric acid). Boiling increased the content of phenolic acids by 3%, while lactic fermentation and roasting decreased the same by 11% and 6%, respectively. Sawicki and Wiczkowski [77] reported a decrease in the content of betalain pigments as a result of boiling and lactic fermentation (respectively by 51–61% and 61–88%). The authors observed that the skin served a protective function, and the pigment loss was lower if the skin was present. The highest antioxidant capacity (measured with ABTS cation radicals) was recorded for skin peeled off fresh beetroots, followed by, in descending order: skin peeled off boiled beetroots, pulp obtained from fresh beetroots, whole fresh beetroots, whole boiled beetroots, beetroots after 14 days of fermentation, beetroots after 7 days of fermentation, and boiled peeled beetroots. Ramirez-Melo et al. [78] compared the impact of the thermosonication process (20 kHz, 2 s impulse, 4 s interval) and classic thermal processing (70 °C/20 min and 80 °C/10 min) on the content of bioactive compounds in beetroot juice. As revealed in the analyses, thermosonication of beetroot juice proved more effective in eliminating bacterial microflora than classic thermal processing. The content of betanin and the antioxidant capacity measured with the ABTS method were both higher in the beetroot juice treated in the process of thermosonication. The control sample contained $253.37 \pm 2.51$ mg betanin/100 mL of juice. After thermosonication and classic thermal processing, the content of betanin was, respectively, $206.44 \pm 2.51$ mg/100 mL and approx. 190 mg/100 mL of juice. The content of phenols after the process of thermosonication, thermal processing (70 °C/20 min and 80 °C/10 min) was, respectively, $202.70 \pm 1.92$, $171.68 \pm 4.60$, and $174.34 \pm 3.43$ mg/100 mL of juice (gallic acid equivalent). Both processes reduced the antioxidant capacity of the juice, with the smallest loss observed in the case of thermosonication. The antioxidant capacity of the control sample was $138.82 \pm 2.30$ µmol TE/100 mL of juice, and of the analytical samples after thermosonication, thermal processing at 70 °C/20 min, and thermal processing at 80 °C/10 min, it was $130.74 \pm 2.79$, $125.74 \pm 4.47$, and $114.17 \pm 1.28$ µmol TE/100 mL of juice, respectively. Bárta et al. [79]

analyzed the impact of boiling (20 min, 100 °C) beetroot extracts from six plant cultivars (Alexis, Betina, Burpee's Golden, Chioggia, D'Egypte, Karkulka) on the content of bioactive compounds and antioxidant capacity. Based on the analyses, it was concluded that boiling reduced the content of betalain pigments in all of the analyzed cultivars, respectively by 3.55, 3.06, 0.59, 0.22, 0.99, and 3.42 mg/g of dry matter. The content of total phenols was also reduced during boiling, with the exception of the Alexis cultivar where a slight increase from $3.25 \pm 0.21$ to $3.49 \pm 0.13$ mg of gallic acid per gram of dry matter was recorded. The antioxidant capacity of beetroot extracts (Alexis, Betina, Burpee's Golden, Chioggia, D'Egypte, Karkulka) measured with the DPPH method was higher, in all analyzed cultivars, after boiling than beforehand. The results recorded prior to boiling were within the range of $4.29 \pm 0.85$ (Chioggia cultivar) and $8.68 \pm 0.90$ (Betina) mg of ascorbic acid/ g d.m., and after boiling between $8.36 \pm 2.77$ (Chioggia) and $17.59 \pm 0.29$ (Betina) mg of ascorbic acid/ g d.m. Dalmau et al. [80] analyzed the impact of convection drying (60 °C, air velocity 2 m/s) and lyophilization (−50 °C, 30 Pa) on the bioavailability of the bioactive compounds present in beetroots. Drying processes decreased the content of phenols by 42% and 29% for convection drying and lyophilization, respectively. The antioxidative capacity was reduced by 66% (convection drying) and 63% (lyophilization). Kaur et al. [81] conducted a study to determine the impact of beetroot blanching on the content of bioactive compounds and antioxidant capacity. The experiment was conducted using the method of steam blanching and cooling (100 °C, 8 min). After blanching, the sample was immediately cooled (15–20 min), then dried and ground. The content of total phenols and betalain pigments was measured in a sample of fresh beetroots and the powder obtained after blanching. The content of betalain pigments after blanching increased to 2469.07 mg/100 g d.m. from $677.34 \pm 0.63$ mg $100\ g^{-1}$ d.m. in the fresh sample. The content of total phenols was also higher after blanching compared to the fresh sample, having increased from $77.88 \pm 0.90$ mg/100 g d.m. to $416.65 \pm 0.58$ mg/100 g d.m. gallic acid equivalent. Antioxidant activity measured with the DPPH method was $47.69 \pm 0.17\%$ in fresh beetroot and $76.54 \pm 0.06\%$ in blanched beetroot powder. The authors also analyzed the impact of the resulting powder's storage on the content of bioactive compounds with only a minimal loss thereof observed. After 3 months in storage, the content of betalain pigments decreased to $1943.38 \pm 0.97$ mg/100 g d.m as compared to the sample immediately after blanching (2469.07 mg/100 g d.m.). In addition, the content of total phenols was reduced to the level of $403.70 \pm 0.50$ mg/100 g d.m from $416.65 \pm 0.58$ mg/100 g d.m. gallic acid equivalent. The antioxidant activity decreased to $71.26 \pm 0.45\%$ after storage. Mohsen et al. [82] studied the impact of the lactic fermentation of beetroot extract on its antioxidant activity. The process of fermentation was conducted with the use of two strains of *Lactobacillus* bacteria (*Lactobacillus plantarum* P108 and *Lactobacillus acidophilus* P110). Based on the conducted analyses, it was concluded that the content of phenolic compounds (TPC) and antioxidant capacity (ABTS method) were both higher in the fermented extracts as compared to unfermented extracts. A comparison between extracts fermented with *Lactobacillus acidophilus* P110 and *Lactobacillus planatrum* P108 revealed that products fermented with the former strain had higher antioxidant capacity and phenol content. The authors concluded that the process of lactic fermentation was beneficial in terms of the content of bioactive ingredients (total phenols) in beetroots.

## 6. Conclusions

Due to the progressive ageing of societies, many regions of the world (e.g., Europe) are currently facing the threat of growing incidences of neurodegenerative diseases, including Alzheimer's disease. The treatment of AD relies primarily on pharmacotherapy. However, for several years now, we have observed a growing interest in nutraceuticals—functional food with a range of health benefits. As posited in numerous scientific papers, beetroot may constitute an interesting alternative in the prevention of neurodegenerative diseases such as AD. Scientists working in various research groups worldwide have analyzed the biochemical content of beetroots in terms of their potential capacity to inhibit the progression

of neurodegenerative diseases. The presence of numerous chemical compounds including betalain pigments, phenols, and saponins was confirmed in the vegetable. Beetroot proves to have huge antioxidant potential. This is due to the presence of not only dyes from the betalain group, but also phenolic compounds such as ferulic, gallic, and vanillic acids. Apart from bioactive compounds, beetroot also contains inorganic compounds such as nitrates. As a result of their transformation, nitric oxide (II) is formed, which not only lowers blood pressure but can also improve blood circulation and oxygenation of the brain. In the elderly, the blood supply to the brain is disrupted. Tissue hypoxia impairs intellectual performance and can cause neurodegenerative diseases. As mentioned, nitric oxide (II) causes dilation of blood vessels and facilitates the flow of oxygen. The presented review discussed said bioactive compounds (pigments, phenols, saponins) and summarized the results of relevant experimental studies conducted with the use of animal models. The overall results are promising, which is why in-depth research on the potential uses of beetroot products in the prevention of diseases such as AD should be continued.

**Author Contributions:** Concept, literature review, writing, J.S.; review and editing, D.S.; review and editing, E.B.-W. All authors have read and agreed to the published version of the manuscript.

**Funding:** This work is supported by the Ministry of Education of the Polish Government (Project No. DWD/5/0020/2021).

**Institutional Review Board Statement:** Not applicable.

**Informed Consent Statement:** Not applicable.

**Data Availability Statement:** Not applicable.

**Conflicts of Interest:** The authors declare no conflict of interest.

## Abbreviations

| | |
|---|---|
| ABTS | 2,2-azinobis (3-ethylbenzothiazoline- 6-sulfonic acid) |
| AchE | Acetylcholinesterase |
| AD | Alzheimer's disease |
| BHT | Butylated hydroxytoluene |
| b.w. | Body weight |
| DMSA | 2,3-dimercaptosuccinic acid |
| DPPH | 2,2-diphenyl-1-picrylhydrazyl |
| d.m. | Dry matter |
| FRAP | Ferric reducing antioxidant power |
| GSH | Glutathione |
| IC50 | Inhibitory concentrations 50% of cells |
| MDA | Malondialdehyde |
| TEAC | Value expressed as Trolox equivalent |

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
