# Peer review of "The Role of Beetroot Ingredients in the Prevention of Alzheimer’s Disease"

_applsci, doi:10.3390/app13021044_

Round 1

Reviewer 1 Report

The manuscript by Szymanski et al reviews the literature describing the potential benefits of beetroots extracts in the prevention of AD.

According to this reviewer the major limit of the manuscript is that it lacks a precise organization and presentation of the available data and therefore it is difficult to read. The title refers to AD, but then the authors describe a whole set of in vitro and in vivo data that are not clearly related to this disorder, but more generally to experimental settings characterized by oxidative stress, inflammation and so on. This happens both in the section dealing with the beet extracts and in those focusing on more specific components. I believe that the manuscript could benefit from a more precise and selective analysis of available data according to the type of assays (in vitro/ in vivo) and the experimental models (general neurodegeneration or AD).

Minor points

-          The manuscript starts with a general description of the main pathogenic pathways involved in the generation of AD. Most of this section is fine, but some paragraphs or affirmations are not clear at all. Describing the Beta-amyloid hypothesis, the authors write ”…in the case of pathological transformations of the peptide…”. What does it mean? The whole paragraph about the pathological involvement of Tau proteins is confusing and unprecise.

-          I find excessive the citation of so many figures of the experimental results. In most cases the general conclusion supported by the data is enough.

-          I think that section 5 is not relevant for this type of review.

-          The references cited in the introduction and in the general description of AD need to be completely changed (in particular the firs 10 references). Some of them are not in English, therefore not available for most of the readers, and there are more relevant and high impact papers that can be cited to support the affirmations included in these sections.

-          In general, I suggest a careful revision of the references, since some of them (n 11, 17, 23, 35), do not fit with the affirmation in the manuscript.

Reviewer 2 Report

I believe that this is an interesting revision analysis towards the prevention of Alzheimer's disease and neurodegeneration in general by beetroot, since this is an important issue especially in the western countries due to the proportion of the elder population.  

I propose the acceptance of this manuscript after some important modifications.

Some issues that need to be addressed and to be taken into account are listed below.

1.    The abstract section should be expanded and reference to some important bioactive compounds that are referred in the main text should be made.

2.  The authors should extensively expand the introduction or discussion section referring in more detail to the oxidative stress and the inflammation that are the major mechanisms for the progression of AD and the neurodegenerative conditions in general, before they analyze the bioactive compounds of beetroots.

Some indicative and recent references that could be added together with others are given:

-        Nutrients 2020, 12, 435

-        Molecules, 2022, 27(23), 8402

-        Medicinal Chemistry, 2021, 17(10), pp. 1086–1103

-        Life Sci 2019, 218, 165-184

3.     The authors should change “Polyphenolic compounds” to Phenolic compounds since many of them described in this text are not polyphenolic.

4.     Figure 1 is very big and the structures are almost enormous. I propose the authors use softwares like chemdraw in order to make their figure more concrete. I also propose to add next to each structure the biological activities that these compounds possess according to their analysis or make a concluding figure at the conclusion describing schematically and cumulatively the targets of these ingredients.

5.     Furthermore, a more extensive conclusion with potential future perspectives should be added, so as the reader to be able to define the future of the application of dietary products like beetroots in the neurodegenerative conditions and AD.

6.    “However, a growing number of interesting studies have been devoted in recent years to the value of a properly selected diet to assist a successful therapy.” The authors should refer to more studies, not only in this case but in also in others in case they state that there is a growing number.

7.     Please revise the text carefully for potential grammatical and typographical errors.

·       e.g. “beetroot preserves can offer viable benefits” what does preserves mean, could the word preserves be omitted.

Round 2

Reviewer 1 Report

The authors partially replied to the criticisms raised by this rewiewer and corrected the manuscript accordingly.